# Strategy for Pre-Clinical Development of Active Targeting MicroRNA Oligonucleotide Therapeutics for Unmet Medical Needs

**DOI:** 10.3390/ijms24087126

**Published:** 2023-04-12

**Authors:** Marc Thibonnier, Sujoy Ghosh

**Affiliations:** 1AptamiR Therapeutics, Inc., Austin, TX 78759, USA; 2Duke-NUS Medical School, Singapore and Pennington Biomedical Research Center, Baton Rouge, LA 70808, USA

**Keywords:** microRNAs, oligonucleotide therapeutics, active targeted delivery, obesity, diabetes, MAFLD, ovarian cancer

## Abstract

We present here an innovative modular and outsourced model of drug research and development for microRNA oligonucleotide therapeutics (miRNA ONTs). This model is being implemented by a biotechnology company, namely AptamiR Therapeutics, in collaboration with Centers of Excellence in Academic Institutions. Our aim is to develop safe, effective and convenient active targeting miRNA ONT agents for the metabolic pandemic of obesity and metabolic-associated fatty liver disease (MAFLD), as well as deadly ovarian cancer.

## 1. Introduction

MicroRNAs (miRNAs) are highly conserved small non-coding RNA molecules that post-transcriptionally regulate gene functions through direct degradation of target mRNAs and/or translational repression. The total number of validated human miRNAs was recently estimated to be around 2300 [1,2]. miRNAs are stable, often display cell-type specificity, can be easily quantified from tissues and body fluids, and play a major role in exosomal cell-to-cell communications [3,4,5,6,7].

miRNAs are convenient diagnostic, prognostic and therapeutics markers of various human diseases (e.g., cardiovascular, metabolic, neurologic, infectious) [8,9,10,11,12] and cancers [13,14,15,16]. In addition, miRNAs are increasingly recognized as modulators of disease pathogeneses and are consequently considered bona fide molecular therapeutic targets [17]. Due to miRNAs’ distinct expression patterns and their ability to target numerous transcripts via a one-to-many relationship, miRNA modulators present a unique opportunity to alter the expression of proteins that cannot be feasibly targeted by small-molecule drug-based approaches [18].

Nucleic acid therapeutics (NATs)/oligonucleotide therapeutics (ONTs) represent a new and growing area of drug discovery and development [19,20]. As of 19 December 2022, ONTs (mainly antisense oligonucleotides (ASOs) and small interfering RNAs (siRNAs)) have been initially approved by regulatory authorities in the US and/or the EU. To facilitate R&D process development of NATs/ONTs, the FDA published in June 2022 a draft guidance titled Clinical Pharmacology Considerations for the Development of Oligonucleotide Therapeutics (https://www.fda.gov/regulatory-information/search-fda-guidance-documents/clinical-pharmacology-considerations-development-oligonucleotide-therapeutics (accessed on 22 June 2022)).

Numerous clinical trials are currently evaluating ONTs for rare disease and cancer treatments [21,22]. Some of these should be approved within the coming years. Moreover, several ONTs have been developed for common prevalent diseases, such as cardiovascular and metabolic disorders. Inclisiran (Leqvio^®^) is an siRNA that targets *PCSK9* approved and marketed for the treatment of hypercholesterolemia [23]. Pelacarsen is an ASO that targets lipoprotein(a), a risk factor for cardiovascular diseases (NCT04023552, ClinicalTrials.gov). Fesomersen is a *Factor XI* ASO, designed to prevent thrombosis ((NCT04534114). ON449 (AZD8233) is an investigational ASO designed to reduce blood cholesterol levels by targeting *PCSK9* (NCT04641299). Zilebesiran (ALN-AGT) is an siRNA that targets liver-expressed *angiotensinogen* to reduce blood pressure in hypertensive patients; it is also currently undergoing phase 2 clinical trials (NCT04936035 and NCT05103332). Additionally, 15 or more additional regulatory approvals are expected in the next 5 years.

The first generation of ONTs to advance into clinical trials incorporated several medicinal chemistry modifications, including the following [24,25,26]:

Phosphorothioate (PS) backbone modifications to reduce nuclease degradation and increase plasma protein binding to facilitate tissues uptake;

Gapmer oligodeoxynucleotides (ODNs) to elicit efficient RNase H cleavage of the target RNA;

Ribose modifications (mainly 2′ position modifications (2′-fluoro, 2′-O-methyl, 2′-O-methoxyethyl) and 2′-4′ locked nucleic acid (LNA)) to enhance stability and specificity.

Although these modifications improved drug potency, they also came with safety issues, such as chirality, hepatotoxicity, renal toxicity, thrombocytopenia, complement activation and immune/hypersensitivity reactions [27,28,29]. Therefore, generation 2 and generation 2.5 ONT drug candidates were designed to avoid toxicity and improve cell penetration and tissue targeting, while also reducing effective therapeutic doses and improving their PK/PD profile, particularly their intra-cellular mean residence time (MRT) [30,31]. To date, at least twelve N-acetylgalactosamine carbohydrate (GalNAc)-conjugated ONTs are in phase II or phase III clinical trials for specific delivery to the liver. Three out of four siRNAs on the market are GalNAc conjugates, including Givorisan/Givlaari^®^, Lumasiran/Oxlumo^®^ and Inclisiran/Leqvio^®^.

Although miRNA-based ONTs, including anti-miR compounds, specific miRNA inhibitors, and miRNA mimics, are being tested in pre-clinical studies, phase 1 clinical trials or phase 2 clinical trials [20,32], regulatory authorities have not yet approved their clinical use beyond investigational drugs. miRNAs’ ability to target multiple genes could be advantageous in managing complex diseases that involve the deregulation of multiple signaling pathways. Advances in active targeting delivery of miRNA-based ONTs should accelerate their transition from bench to bedside in the coming years [33]. To improve their PK/PD and safety profile, new chemical modifications have been introduced into the new generations of ONTs. For instance, the novel class of gamma peptide nucleic acid (γPNA) compounds are water-soluble and charge-neutral. Additionally, they neither aggregate nor adhere to surfaces or other macromolecules in a nonspecific manner and they adopt a right-handed helical motif, hybridize to DNA or RNA with an unusually high affinity and sequence specificity [34,35,36]. Each γ modification stabilizes a PNA–RNA duplex by 5 °C. Compared to LNAs and 2′-OMe oligonucleotides, PNAs are shown to be more potent inhibitors of targeted miRNA activity [37].

Although in silico tools can predict the binding of a given miRNA to many target mRNAs, its true biological effects depend on multiple factors, including the tissue expression level of the miRNA, availability of the Argonaute-2 protein (AGO2) and the expression and abundance of the miRNA target genes of interest.

There is mounting experimental evidence that miRNAs can be used as diagnostic/prognostic markers and potential therapeutic targets in various cancers [38,39]. miRNA dysregulation is involved in epithelial–mesenchymal transition (EMT), invasion, proliferation, migration, metastasis, angiogenesis, immune responses and drug resistance of various cancers [40,41,42,43]. Restoring the expression of tumor-suppressor miRNAs or inhibiting overexpressed oncogenic miRNAs (oncomiRs) are promising strategies for targeted cancer therapies. Recently, a peptide nucleic-acid-based antisense has been shown to be a potential new drug candidate for pancreatic cancer [44]. Accumulating evidence suggests that exosomal miRNAs are relevant players in dynamic crosstalk among cancerous, immune, and stromal cells in establishing the tumorigenic microenvironment [45]. In addition, they sustain metastatic niche formation at distant sites.

At the inception of AptamiR Therapeutics, we implemented an R&D strategy of **small throughput–high yield** for the development of miRNA-based ONTs. Instead of relying on brute force drug screening (so called high throughput–low yield screening), we used an innovative model of modular and outsourced drug R&D following the “learn and confirm” strategy championed by Dr. Lewis Sheiner at UCSF [46]. Because of the fast pace of discoveries and publications in the miRNA field and the conveniently expanding commercial availability of tools and reagents required to conduct this kind of R&D activity, AptamiR Therapeutics, Inc. uses a **modular parallel and iterative approach,** rather than a classical **serial** one (Figure 1)**.**

Specifically, we focused on miRNAs as therapeutic agents for the treatment of metabolic pandemics, namely obesity, metabolic (dysfunction)-associated fatty liver disease (MAFLD) and, more recently, ovarian cancer.

## 2. Results

### 2.1. Selection of MicroRNAs Involved in Adipose Tissue Functions to Treat Metabolic Pandemics

#### 2.1.1. Background and Goal

Obesity is a worldwide pandemic, affecting 1/3 of the world population, including children. Its medical, economic and social burdens are significant and growing. Current pharmacotherapy rates for obesity are very low because of limited efficacy, significant side effects, adverse events, restricted access and removal of previously approved drugs. Novel therapies of obesity dramatically improve the lives of millions of subjects, diminish healthcare costs and reducing the death toll linked to obesity-associated cancers and cardiovascular diseases [47,48]. Metabolic (dysfunction)-associated fatty liver disease (MAFLD) is a proposed new terminology that more accurately reflects liver pathogenesis and can help patient stratification for the management of fatty liver disease [49]. MAFLD affects 25% of the global adult population, especially obese patients. There is currently no approved drug to treat MAFLD. MAFLD is the major cause of chronic liver disease, has become the most frequent reason for liver transplantation, and it is associated with substantial morbidity and mortality. The relationships between fat accumulation, inflammation and necrosis resulting in lipotoxicity, dyslipidemia, insulin resistance, diabetes, liver steatosis, inflammation and fibrosis are shown in Figure 2, adapted from [50].

Our goal is to develop a novel, safe, effective and convenient therapy for metabolic pandemics, such as obesity and MAFLD. However, obesity and MAFLD are multifactorial diseases that cannot be easily controlled by classical therapeutic agents based on the classical one drug, one target mechanism of action. As microRNAs play diverse roles in obesity and metabolic diseases [51,52,53], as well as regulate gene functions through a one-to-many relationship, our therapeutic strategy is based on the three following innovative principles:

Targeting fat-storing white adipocytes to transform them into fat-burning adipocytes (“browning effect”), instead of altering brain functions, such as appetite/satiety or reducing food intake and absorption (the focus of most pharmacological approaches with significant side effects).

Focusing on miRNA-based ONTs that simultaneously modulate many target genes involved in lipid oxidation, energy expenditure and chronic inflammation (one drug, multiple targets concept) and are well suited for complex and prevalent diseases, such as obesity and MAFLD.

Developing a unique delivery platform to actively target human adipocytes.

Through these efforts, we expect to deliver on our end goal, which is to help patients live longer and healthier lives, while reducing healthcare expenditure.

#### 2.1.2. Strategy

We initiated a search for miRNAs that are unique, conserved across species, universal, abundantly expressed in metabolic tissues/organs and modulate metabolic pathways, especially thermogenesis. This discovery phase of the project was conducted via in silico mining of publicly available datasets, presenting miRNAs as putative regulators of candidate genes.

#### 2.1.3. In Silico Search

Our initial in silico analysis identified putative miRNAs from a known pool of about 2000 miRNAs and a digitally curated list of 721 genes involved in lipid metabolism, oxidative phosphorylation, mitochondrial functions, respiratory cycle, browning of adipocytes and energy expenditure (AptamiR 721 genes were selected using eight publicly available in silico tools, namely BioCarta; Database for Annotation, Visualization and Integrated Discovery (DAVID); GeneOntology; Gene Set Enrichment Analysis (GSEA); Kyoto Encyclopedia of Genes and Genomes (KEGG); PubGene; Reactome; and STRING). We utilized 34 in silico miRNA target prediction tools and our own proprietary in silico meta-tool (R-AptamiR) to identify 200 miRNAs that potentially bind to these target metabolic genes. A metabolic target of particular interest is the mitochondrial uncoupling protein UCP1, which increases thermogenesis in adipose tissues. The human *UCP1* gene structure is notable for a high degree of methylation (“CG islands”) in its promoter region. Methylation of CG islands within gene promoters can lead to their silencing [54]. The human lysine (K)-specific demethylase 3A (KDM3A) is critically important in regulating the expression of metabolic genes and obesity resistance [55]. Using microarray technology, Zhang et al. demonstrated that a significant number of the genes involved in PPAR signaling and fatty acid oxidation (e.g., *PPARA, ACADM, ACADL, ACADVL, AQP7*) were down-regulated in response to *KDM3A* knockout [55]. KDM3A directly regulates peroxisome proliferator-activated receptor alpha (PPARA) and UCP1 expression [56]. After a few “learn and confirm” rounds between in silico, in vitro and in vivo experiments, we identified a subset of miRNA targets from the original 200. We found that the human *KDM3A* 3′ UTR 29–35 region is a conserved target for hsa-miR-22-3p, as shown in Figure 3.

The expression of *KDM3A* is widely distributed across human tissues and organs with low tissue specificity of 0.29 (www.proteinatlas.org/ (accessed on 20 February 2023)).

#### 2.1.4. Validation of miR-22-3p as a Metabolic Target

Using various in silico, in vitro and in vivo tools, we demonstrated that miR-22-3p is an excellent metabolic target that modulates several genes, as illustrated below using the metaMIR tool (http://rna.informatik.uni-freiburg.de (accessed on 20 February 2023)), which ranks miRNAs in relation to gene networks (Table 1).

As these genes are required for normal metabolic functions and miR-22-3p is likely to induce degradation or reduce the translation of these genes, a strategy involving antagonizing the functions of miR-22-3p constitutes a viable therapeutic route to the amelioration of metabolic disorders. We further used the protein–protein interaction functional enrichment analysis tool STRING (https://string-db.org (accessed on 3 July 2021)) [57] to illustrate the various interactions within a network of 34 proteins related to miR-22 (Figure 4).

The human tissue/organ distribution of hsa-miR-22-3p was examined using the TissueAtlas2 program (https://www.ccb.uni-saarland.de/tissueatlas2 (accessed on 20 February 2023)) [58]. The tissue specificity index (TSI) gives each specific non-coding RNA molecule a numeric value on a scale from 0 to 1, with 1 meaning that the expression of the molecule was detected in only one specific tissue and 0 meaning that the expression of the molecule was detected in all tissues. The TSI for hsa-miR-22-3p was 0.855, indicating a high tissue-specific expression pattern, especially in the adipocytes, the myocardium and skeletal muscle (red arrows, Figure 5).

#### 2.1.5. In Vitro and In Vivo Validation of miR-22-3p Antagonism

We then proceeded to explore the metabolic effects of miR-22-3p antagomirs in vitro in primary cultures of human adipocytes and in vivo in mice. The metabolic and energetic benefits of first-generation miR-22-3p antagomirs were summarized in two peer-reviewed articles published in 2020 [59,60]. In vivo proof of concept of miR-22-3p inhibition in mice was performed in the mouse model of diet-induced obesity (DIO) in C57BL/6J male mice. Mice of various ages were allocated to normal chow (10% fat) or a 60% high-fat diet and were treated for up to 12 weeks with a miR-22-3p antagomir or saline. We consistently observed in the presence of miR-22-3p antagomir treatment a reduction in body weight and fat mass without alteration of lean mass; improvement in glucose, insulin sensitivity and lipid profile; increase in thermogenesis; and no modification of food intake or body temperature.

Treatment for 12 weeks with miR-22-3p antagomir APT-110 produced a marked reduction in fatty infiltration of the liver (Figure 6) [60].

#### 2.1.6. Safety Assessment

Pre-IND toxicology and safety studies of first-generation miR-22-3p antagomir APT-110 (a “naked” single-stranded 18 mer miR-22-3p antagomir containing PS and LNA modifications) were completed in mice and non-human primates according to FDA guidance. In mice, APT-110 administered subcutaneously at 15, 60, and 240 mg/kg/dose on days 1, 3, 5, 8, 15, 22 and 29 was well tolerated and associated with atrophy of adipose tissues at all dose levels. In cynomolgus monkeys, APT-110 was administered subcutaneously at 3.75, 15, 37.5 and 60 mg/kg/dose on days 1, 3, 5, 8, 15, 22 and 29 of the study. Transient activation of blood platelets and the complement pathway was observed right after administration of supra-therapeutic doses. Kidney and liver histologic alterations were also noted. Due to these observations, our first-generation compound was not advanced further.

#### 2.1.7. Design of Generation 2.5 ONTs for Active Targeted Delivery to Metabolic Tissues/Organs

Our new generation 2.5 of targeting ONTs was designed to conduct the following:Eliminate potential toxicities by replacing PS and LNA modifications with a gamma PNA backbone;Maintain resistance to nucleases and proteases/peptidases;Avoid chirality;Limit binding to serum proteins;Optimize/simplify chemical synthesis;Conjugate ONT to a fatty acid or a short peptide for enhanced targeted delivery to adipocytes of a greatly reduced effective dose with an extended duration of action (mean residence time).

#### 2.1.8. Selection of the Membrane Fatty Acid Translocase (FAT) Transporter for Active Targeted Delivery of ONTs to Adipocytes and Metabolic Organs

The membrane transporter FAT/CD36/SCARB3 is the main route of uptake by adipose tissues of long-chain fatty acids, as well as short peptides, such as hexarelin, prohibitin and thrombospondin peptide-1 [61,62]. FAT is significantly expressed in cells and tissues sensitive to metabolic dysfunctions, such as adipocytes, hepatocytes, skeletal and cardiac myocytes, pancreatic β-cells, kidney glomeruli and tubules cells, monocytes and macrophages. As the average obese male patient weighs around 200 lbs, of which 40% is adipose tissue, there is a huge amount of FAT available at the surface of the adipose tissues to transport inside the adipocytes our new generation 2.5 of miR-22-3p antagomirs coupled with a fatty acid or peptide. We compared the mRNA and protein level expression of FAT across human tissues to that of asialoglycoprotein receptor 1 (ASGR1), which has been successfully targeted for the preferred delivery of ONTs to the liver (Figure 7). This comparison illustrates the rationale for targeting FAT for the delivery of ONTs to metabolic organs.

#### 2.1.9. In Silico Modeling of Generation 2.5 ONTs

We then focused on significantly reducing the effective dose of ONTs to obtain a greatly improved safety and PK/PD profile, especially in terms of the mean residence time inside the targeted cells. Pr. Pengyu Ren, Ph.D. and his associates at the Department of Biomedical Engineering at the University of Texas at Austin performed high-performance molecular dynamics modeling on graphics processing units of generation 2.5 miR-22-3p antagomirs, coupled with a fatty acid of increasing length from C16 palmitic acid to C22:6 docosahexaenoic acid and C32:6 dotriacontahexaenoic acid (Figure 8) [63,64,65].

To further refine our ability to accurately deliver miRNA antagonists to target tissues, we also studied the coupling of generation 2.5 miR-22-3p antagomirs to the peptide hexarelin. Hexarelin (His-D-2MeTrp-Ala-Trp-D-Phe-Lys-NH2) is a stable analog of growth-hormone-releasing peptide 6 and is a high-affinity ligand for FAT [66]. The interaction of hexarelin with FAT promotes the transcriptional activation of nuclear receptor PPARγ and genes involved in metabolism and thermogenesis [67,68]. Based on these findings, we are also developing generation 2.5 miR-22-3p antagomirs coupled with Hexarelin for active targeting of adipocytes (Figure 9).

Together with these optimization studies, we are currently conducting in vitro tests of primary cultures from human adipocytes and in vivo tests of animal models with obesity and MAFLD to understand the metabolic benefits of our generation 2.5 miR-22-3p antagomirs.

### 2.2. Selection and Targeting of MicroRNAs Involved in Ovarian Cancer Development and Spread

#### 2.2.1. Background and Goal

Cancer epidemiologic and mortality surveys report that ≥300,000 women are diagnosed with ovarian cancer (OC) worldwide and ≥200,000 succumb to OC every year [69,70]. Most patients are diagnosed with advanced OC at stages III or IV. High-grade serous OC (HGSOC) is the most common and deadliest type of OC. Survival of 5 years is only 30% in HGSOC and 18% for patients diagnosed with stage IV tumors. Presently, debulking cytoreductive surgery is the gold standard for the treatment of OC, along with platinum-based chemotherapy regimens. For patients who become platinum resistant, few options are available, and the efficacy of these regimens is limited. Regardless of treatment options, survival rates of OC have barely improved over the years due to lack of specific symptoms and early screening/diagnostic tools, as well as the high rates of drug resistance and cancer relapse. Due to its clinical, biological and molecular complexity, OC is a tumor that is difficult to treat and lacks a clear driver mutation. Targeted therapies are now a strong focus for OC [71,72].

There is now ample evidence that miRNAs play many roles in OC, its tumor microenvironment and resistance to treatments [73,74,75,76,77,78]. Several membrane receptors often overexpressed in OC cells have emerged as potential targets for receptor-mediated therapies [79]. Given the roles of miRNAs in various cellular pathways, including cell survival and differentiation, targeting miRNAs could be a viable approach for the treatment of human cancers, such as OC, through the inhibition and/or stimulation of miRNAs [80]. Our end goal is to develop safe, effective and convenient miRNA-based therapies to cure OC.

#### 2.2.2. Strategy

Building on our knowledge of generation 2.5 miRNA ONTs and targeted delivery to specific cell types, we decided to develop microRNA-based targeting ONTs that can simultaneously modulate several target genes involved in OC and its tumor microenvironment and spreading. We found that our modular parallel and iterative approach could be quickly adapted from one therapeutic area to another.

#### 2.2.3. In Silico Search

In collaboration with the Centre for Computational Biology and Program in Cardiovascular & Metabolic Disorders at Duke-NUS Medical School, Singapore, we used various bioinformatics tools to identify relevant miRNAs involved in OC, some of which are listed in Table 2.

Based on the bioinformatics analyses, we found 441 miRNAs and 1003 potential target genes involved in the proliferation, migration, apoptosis, invasion, metastases, differentiation, epithelial–mesenchymal transition activation and chemoresistance of OC. Of these, 14 miRNAs were associated with >40 target genes, 29 miRNAs with 30–39 genes, 71 miRNAs with 20–29 genes and 244 miRNAs with 5–19 genes. We performed pathway and Gene Ontology biological process over-representation analysis of the 1003 target genes using Webgestalt (http://www.webgestalt.org/ (accessed on 7 November 2022)) and found that genes were over-represented in pathways related to p53, Ras and PI3-kinase signaling, among others (Figure 10).

We then built networks of proteins involved in OC as shown in Table 3.

We also built miRNA-mRNA networks. From these networks, 114 miRNAs were shown to interact with 20 to 59 potential targets, as shown in Table 4.

Examples of miRNA-mRNA networks are shown in Figure 11. The two main categories of miRNAs were revealed as follows: one category (e.g., miR-506-3p, miR-204-5p, miR-23a-3p, miR30c-5p, miR-766-3p, miR-181b-5p, miR-214-3p and miR-664a-3p) seems to interact with a specific set of target genes; and the other category (e.g.,miR-34a-5p, miR-29a-3p, miR-92a-3p, miR-145-5p, miR-16-5p, miR-93-5p, miR-15b5p, miR-182-5p and miR-200b-3p) seems to share common target genes.

#### 2.2.4. Active Targeted Delivery of Generation 2.5 MiRNA-Based ONTs to Ovarian Cancer Cells via Folic Acid Receptor Alpha (FOLR1)

Building on our knowledge of targeted delivery to specific cell types, as well as learning from the recent development and regulatory approval of antibody drug conjugates (ADCs) and bispecific antibodies (bsAbs) for the treatment of cancers [81,82,83,84,85,86], we looked for a validated target to deliver ONTs to primary OC cells. We selected folic acid receptor alpha (FOLR1), a glycosylphosphatidylinositol (GPI)-anchored cell-surface glycoprotein that is highly expressed at the surface of epithelial ovarian cancer cells [87,88] (Figure 12).

Gene-expression profiling of FOLR1 across tumor samples and paired normal tissues (http://gepia.cancer-pku.cn (accessed on17 February 2023)) shows it to be preferentially expressed in ovarian tumors (Figure 13).

Up to 90% of OC, especially the HGSOC type, overexpress FOLR1 [87,89], and FOLR1 expression is closely associated with the severity of OC (Figure 14) [79,90].

Several FOLR1-targeted therapeutics are currently in late phase clinical trials [91]. Consequently, using our learnings from targeting miRNA therapeutics for obesity and MAFLD, we are now developing ONT candidates conjugated to a fatty acid or a short peptide for enhanced targeted delivery to primary OC cells.

#### 2.2.5. Active Targeted Delivery of Generation 2.5 miRNA-Based ONTs to the Adipocyte-Rich OC Tumor Microenvironment via FAT and FABP4 Transporters

We selected the fatty acid transporters FAT and FABP4, which are highly expressed at the surface of adipocytes (www.proteinatlas.org (accessed on 17 February 2023)), because the adipose-rich omentum microenvironment (cancer-associated adipocytes, CAAs) plays important roles in the spreading and resistance to treatments of ovarian cancer [92]. As recently reviewed by Motohara et al. [92], adipocytes present in the OC tumor microenvironment could augment cancer cell survival, spreading and resistance to chemotherapy via several mechanisms as shown in Figure 15.

#### 2.2.6. In Vitro Testing of Generation 2.5 Candidate miRNA Agomirs and Antagomirs Will Be Conducted in Human Cells in Culture including the Following: 

Epithelial ovarian cancer cell lines, namely SKOV3, SKOV3/CDDP, PA1, CAOV3, SW626, ES-2 and HO-8910;

Negative control cell lines and other cancer cell lines, namely HepG2 (liver) and A-549 VIM RFP (lung cancer);

Primary cultures of human adipocytes.

Cellular high-content Imaging will be performed using Phenovista (www.phenovista.com (to be performed)). Gene profiling will be conducted using the Nanostring PanCancer IO 360 Gene-Expression Panel (770 unique gene-expression panels) at the single-cell (CosMx SMI) and multi-cellular (GeoMix DSP) levels (www.nanostring.com (to be performed)). Direct identification of miRNA targets will be conducted using ECLIPSEBIO miR-eCLIP technology (www.eclipsebio.com (to be performed)).

Finally, in vivo testing of selected candidate targeting miRNA agomirs and antagomirs will be conducted in orthotopic and patient-derived tumor xenograft (PDX) mouse models of ovarian cancer.

## 3. Discussion and Conclusions

miRNA-based treatments provide a new frontier in the next generation of therapeutic development and are expected to significantly impact the landscape of disease management. By departing from the traditional “one drug, one target” approach to a “one drug, many targets” paradigm, miRNA therapeutics are expected to reduce polypharmacy and provide significant gains in drug efficacy, safety and patient compliance. Of course, issues surrounding the biophysical properties of miRNA-based treatments, such as in vivo stability, delivery and potential side effects due to immune activation, need to be addressed to enable safer and more efficacious biologics. In our own case, we identified miR-22-3p as a promising candidate for the treatment of metabolic disorders associated with obesity and MAFLD and demonstrated the efficacy of mir-22-3p antagomirs both in vitro in human cells and in vivo pre-clinical animal models. Currently, we are using computational modeling, bioinformatics and experimental approaches to develop generation 2.5 miR-22-3p antagomirs with improved efficacy and reduced side effects for the treatment of the aforementioned metabolic disorders. Furthermore, we are currently exploring the application of miRNA agomirs and antagomirs for the treatment of ovarian cancers where unmet medical needs are quite large.

## 4. Materials and Methods

### 4.1. In Vivo Experiments

Diet-induced obesity (DIO) male C57BL/6J mice purchased from the Jackson Laboratory (Bar Harbor, ME, USA) were housed at 24–26 °C with lights turned on at 08:00 and off at 20:00. They were fed at weaning on chow (Beekay rat and mouse diet 1). From six weeks of age, they were started on a 60% high-fat diet (Research Diet D12492). At the age of 12 weeks, the mice were allocated to treatment groups so that the mean and standard deviation for body weight, glucose and insulin were similar across the groups (12 animals per group, 2 animals of the same group per cage). After acclimation for 2 weeks, the mice were administered single subcutaneous injections of saline or the APT-110 miR-22-3p antagomir (15 mg/kg) in the left inguinal fat pad (injections on days 0, 2 and 4 of week 1 of treatment, then once a week for a total of 12 weeks) while they remained on the 60% high-fat diet.

Body weights were measured weekly. Food consumption per cage was measured daily. Blood samples were collected from the cut tip of the tail after the application of lignocaine gel during the in-life phase of the study. For plasma preparation, blood was collected in EDTA-coated microvettes for the measurement of plasma analytes and stored on ice, followed by centrifugation at ~5000× *g* for 5 min. The resulting plasma was stored at −80 °C until required. Multiple freeze/thaw cycles were avoided. Blood glucose concentration was analyzed as previously reported [93]. Plasma insulin (Cat #: 90080; Crystal Chem, Downers Grove, IL, USA), leptin (Cat #: 90030; Chrystal Chem), adiponectin (Cat # 47-ADPMS-E01, Alpco Diagnostics, Salem, NH, USA), NEFA (Cat # NEFA-HR(2); Wako Diagnostics, Mountain View, CA, USA), ALT (Cat # AL1205, Randox, Crumlin, UK), AST (Cat # AS1202, Randox), total cholesterol (Cat # CH200, Randox) and triglycerides (Cat # TR210, Randox) were measured as per the manufacturers’ recommendations.

Oral glucose tolerance test (OGTT) was performed as follows: six hours prior to the start of the glucose tolerance test (09h00), food was removed, and animals were given clean cages. Mice were dosed with glucose at T = 0 min. Glucose was dosed by oral gavage at a dose of 2.5 g/kg p.o. Blood samples were taken for the analysis of glucose concentration at −30, 0, 30, 60, 120 and 180 min relative to glucose administration. Blood samples were also taken at −30 and +30 min for insulin analysis. Food was returned at the end of the tolerance test.

Mice energy expenditure (EE) was measured by open-circuit calorimetry with the animals in their home cages [93,94]. The physical activity of the mice was recorded while the mice were kept in their original cages. Recordings were taken using an infrared recorder linked to a laptop. Recordings were made on the hour every hour from 7 pm until 8 am. Each recording lasted 10 min. Analysis of activity was conducted by virtually drawing two lines across each cage (thus dividing them into three equal parts). Recordings were analyzed by eye and the number of times a line was broken by a mouse in each 10 min segment was scored.

Body fat and lean content were measured using a Minispec LF90II Nuclear Magnetic Resonance (Bruker Corporation, Fremont, CA, USA). The mice were gently restrained, sufficient to keep them quiescent during this non-invasive technique.

At the end of the study, liver, heart, inguinal, perirenal, epididymal and subscapular fat samples were collected, weighed, then frozen for future gene-expression analysis; otherwise, they were placed in a 10% neutral buffered formalin solution, washed in PBS, pH 7.4 and transferred to 70% ethanol for subsequent processing for histologic analyses. Spleens were weighed and discarded. Blood was collected and processed into serum or plasma aliquots at the time of necropsy by cardiocentesis.

### 4.2. MiR-22 Antagomir

The miR-22-3p antagomir used in this study was designed by AptamiR Therapeutics, Inc. and custom synthesized (US Patent 62/329,537 on “Inhibition of mir-22 miRNA by APT-110”, initially published on 2 November 2017, WO2017/187426 A1). APT-110 is an 18 mer single-stranded oligonucleotide complementary to nucleotides 2 to 19 of miR-22-3p with phosphorothioate, locked nucleic acid, 2′0-Methyl, DNA and 5-methyl-Cytosine modifications. For subcutaneous dosing in the left inguinal area, APT-110 was prepared in saline (0.9 % NaCl). Control animals received the saline vehicle alone (5 mL/kg).

### 4.3. Imaging Analyses

Measurement of cells’ perimeter and area utilized the ImageJ image processing program developed by the NIH (https://imagej.nih.gov/ (accessed on 22 June 2020)).

### 4.4. RNA Sequencing Analyses

RNA sequencing was performed at the Genome Sequencing and Analysis Facility at the University of Texas in Austin on an Illumina HiSeq 4000 system following the manufacturer’s protocol. RNA data analyses were performed by Dr. Sujoy Ghosh at Duke-NUS Medical School, Singapore. Differentially expressed genes from fat (nominal *p* < 0.01) and liver (nominal *p* < 0.001) gene-expression data were subjected to pathway over-representation analysis via the Enrichr tool [95] (http://amp.pharm.mssm.edu/Enrichr/ (accessed on 15 November 2019)). A total of 1257 and 1495 genes were analyzed from fat and liver, respectively. Pathway enrichment was further investigated via the KEGG pathway database [96] (www.genome.jp/kegg/pathway.html (accessed on 15 November 2019)) and the WikiPathways database [97] (wikipathways.org). Pathways with a false-discovery rate < 0.1 were considered significantly enriched for differentially expressed genes. A subset of significantly enriched KEGG pathways (adj. *p* < 1.5 × 10^−5^) were further visualized via mean-average (MA) plots.

### 4.5. Statistical Analyses

Results given in the text and data points in the figures are shown as mean ± SEM. Statistical analysis used ANOVA and Student’s *t*-test, unless non-parametric tests were selected, based on data distribution (GraphPad Prism 8.4).

### 4.6. Studies Approval

The animal studies were performed according to IACUC-approved protocols and in compliance with the Guide for the Care and Use of Laboratory Animals (National Research Council, 2011) in OLAW-assured and AAALAC-accredited facilities at the Jackson Laboratory, Sacramento, CA (Study ID 40239, 17 April 2015), the University of Buckingham, UK (Study Bu15/030, 30 July 2015), the Drug Dynamics Institute at the University of Texas College of Pharmacy, Austin, TX (IACUC protocol AUP-2015-00125, 10 February 2016) and Aptuit S.r.l., Verona, Italy (Study VPT4074, 22 March 2016).

## Figures and Tables

**Figure 1 ijms-24-07126-f001:**
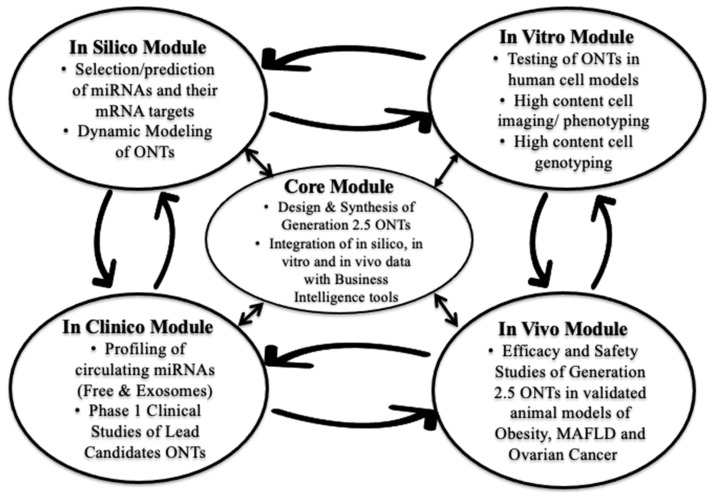
AptamiR Therapeutics’ Modular Parallel and Iterative Strategy of Drug Discovery and Development.

**Figure 2 ijms-24-07126-f002:**
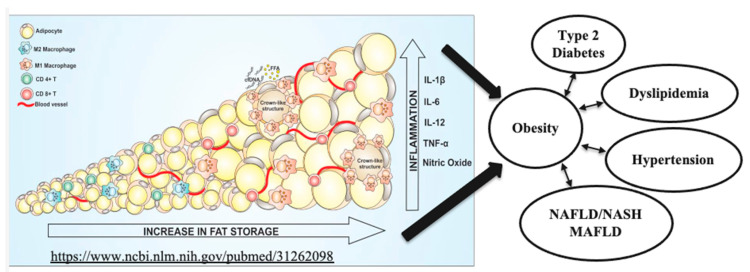
Medical consequences of adipose tissue hypertrophy, hyperplasia, inflammation and necrosis (adapted from Reference [50], an open-access article distributed under the terms and conditions of the Creative Commons Attribution (CC BY) license (http://creativecommons.org/licenses/by/4.0/ (https://www.ncbi.nlm.nih.gov/pubmed/31262098 accessed on 20 February 2023)).

**Figure 3 ijms-24-07126-f003:**
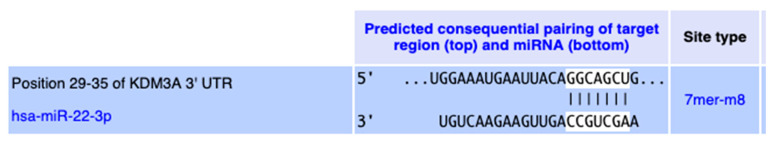
Conserved interaction between hsa-miR-22-3p and the 3′-UTR region of the *KDM3A* gene (TargetScanHuman 8.0, www.targetscan.org/vert_80/ (accessed on 20 February 2023)).

**Figure 4 ijms-24-07126-f004:**
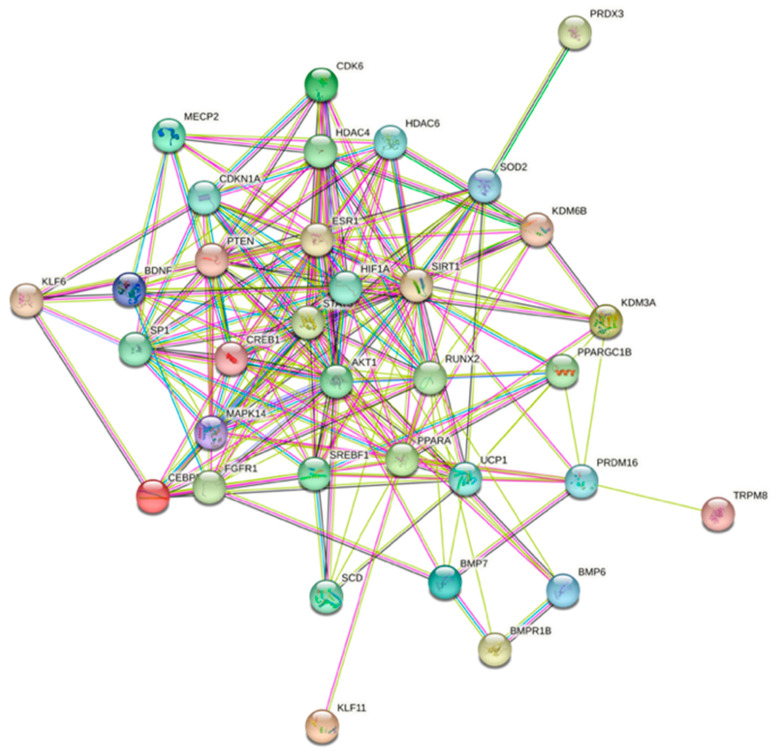
Prediction of protein–protein interaction networks related to miR-22 using the protein–protein interaction-networks functional enrichment analysis tool String (https://string-db.org (accessed on 3 July 2021)).

**Figure 5 ijms-24-07126-f005:**
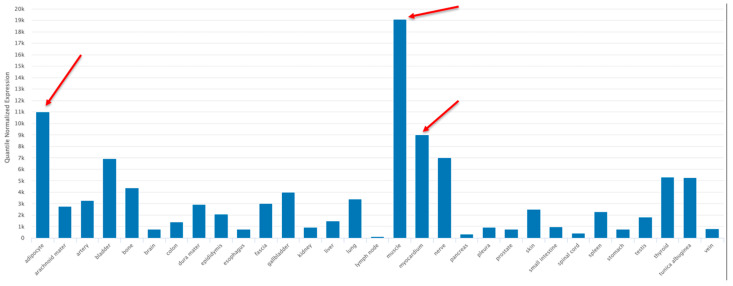
Tissue/organ distribution of miR-22-3p using the TissueAtlas2 program (https://www.ccb.uni-saarland.de/tissueatlas2 (accessed on 20 February 2023)) [58], an Open Access article distributed under the terms of the Creative Commons Attribution-NonCommercial License (https://creativecommons.org/licenses/by-nc/4.0/, accessed on 20 February 2023). Adipocyte, myocardium and skeletal muscle samples are identified by red arrows.

**Figure 6 ijms-24-07126-f006:**
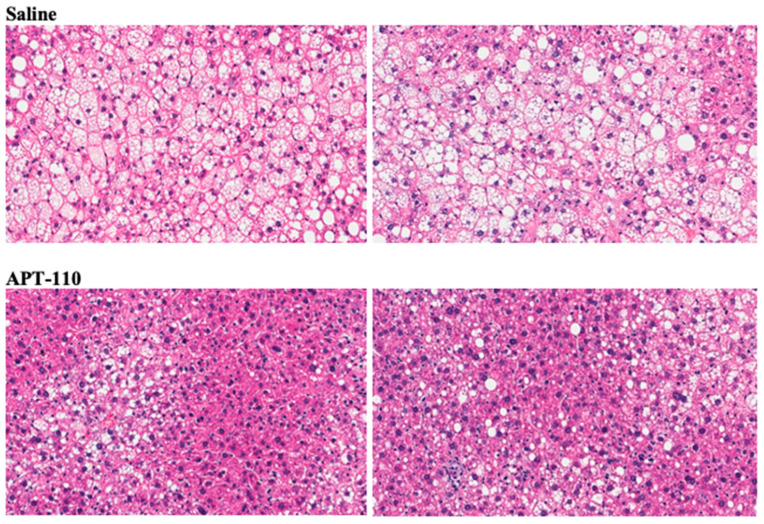
Histologic appearance of livers (H&E staining) at the end of 12 weeks of treatment in mice receiving SC injections of saline or the APT-110 miR-22-3p inhibitor (two samples from each group are shown), reprinted from Reference [60], an open-access article distributed in accordance with the Creative Commons Attribution Non-Commercial (CC BY-NC 4.0) license.

**Figure 7 ijms-24-07126-f007:**
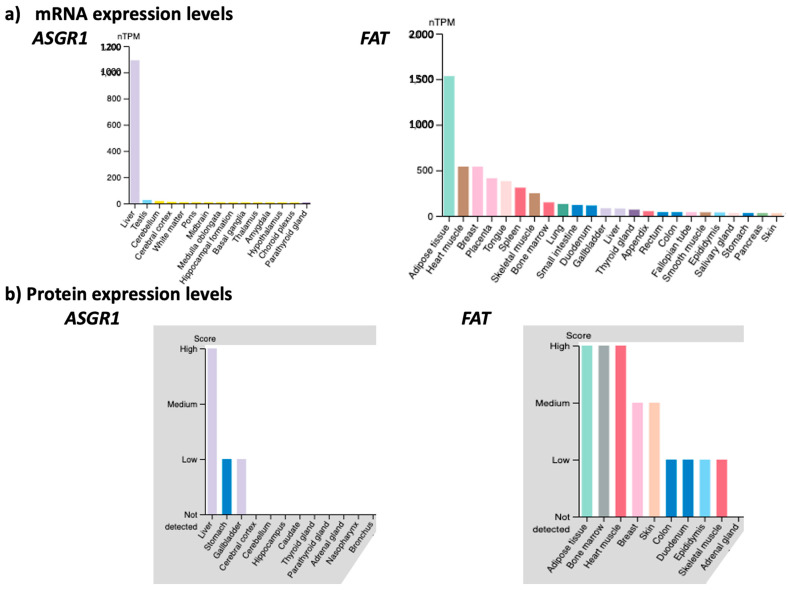
Comparison of mRNA (**Panels a**) and protein (**Panels b**) expression levels in various tissues and organs of the asialoglycoprotein receptor 1 (ASGR1) and the fatty acid translocase (FAT) membrane transporter (nTPM = normalized transcript per million) (www.proteinatlas.org (accessed on 27 March 2023)). The figures are shown in “Expression Level” mode in decreasing order from left to right. The tissues/organs with little or no expression are not shown for sake of readability.

**Figure 8 ijms-24-07126-f008:**
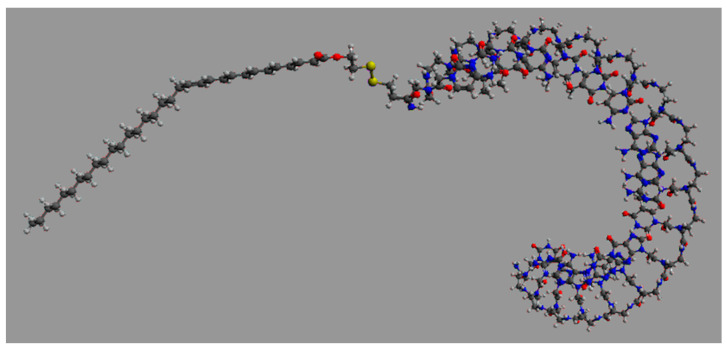
Graphic representation of an 18 mer miR-22-3p antagomir with a PNA (Pna) backbone coupled with C32:6 dotriacontahexaenoic fatty acid (5′-C32-S-S-PnaC-PnaT-PnaT-PnaC-PnaT-PnaT-PnaC-PnaA-PnaA-PnaC-PnaT-PnaG-PnaG-PnaC-PnaA-PnaG-PnaC-PnaT-3′).

**Figure 9 ijms-24-07126-f009:**
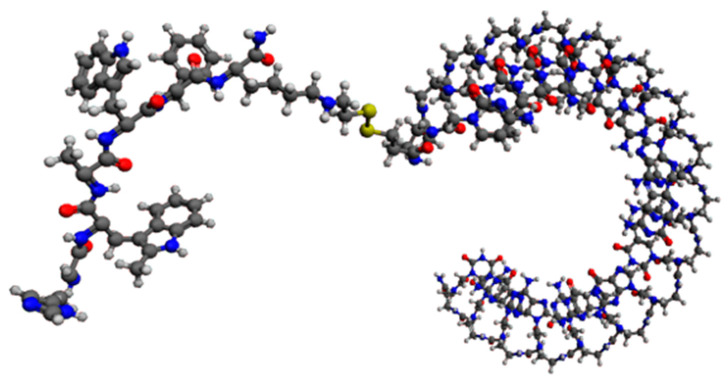
Graphic representation of an 18 mer miR-22-3p antagomir with a PNA backbone coupled with the hexapeptide hexarelin (5′-Hex-S-S-PnaC-PnaT-PnaT-PnaC-PnaT-PnaT-PnaC-PnaA-PnaA-PnaC-PnaT-PnaG-PnaG-PnaC-PnaA-PnaG-PnaC-PnaT-3′).

**Figure 10 ijms-24-07126-f010:**
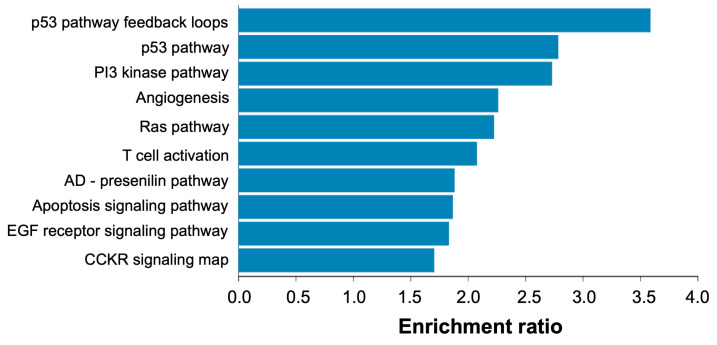
Pathway over-representation analysis of candidate miRNA target genes relevant to ovarian cancer. Analysis was performed via Webgestalt (http://www.webgestalt.org/ (accessed on 7 November 2022)) against Panther pathways and using whole human genome as background. The x-axis refers to the enrichment ratio for each pathway (all pathways were significantly enriched at FDR < 5%).

**Figure 11 ijms-24-07126-f011:**
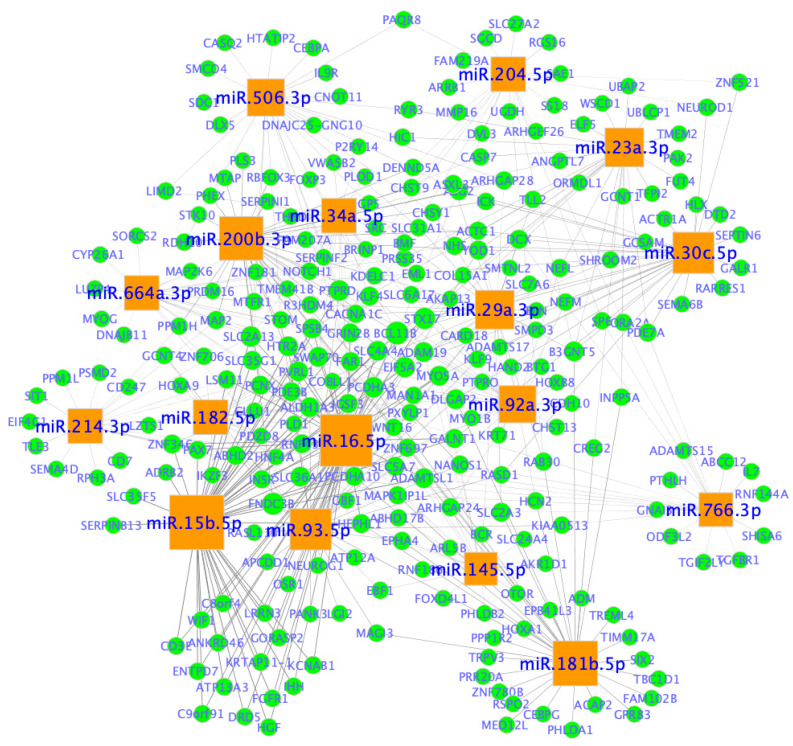
miRNA-mRNA networks in ovarian cancer. Network nodes represent miRNAs and genes (mRNAs), whereas edges represent an association between a miRNA and a gene. miRNAs are represented by orange squares and genes are shown in blue circles. miRNA node size is proportional to the number of genes it is predicted to interact with. Network was created in Cytoscape (https://cytoscape.org/ (accessed on 20 January 2023)).

**Figure 12 ijms-24-07126-f012:**
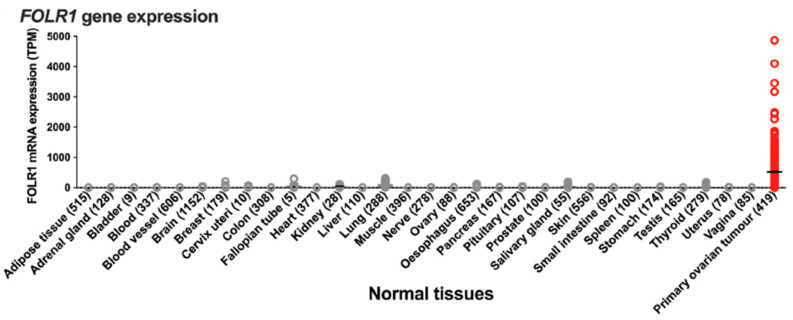
Tissue/organ distribution of FOLR1 in normal tissues and ovarian cancer, reprinted from [87], an article licensed under Creative Commons Attribution 4.0 International License.

**Figure 13 ijms-24-07126-f013:**
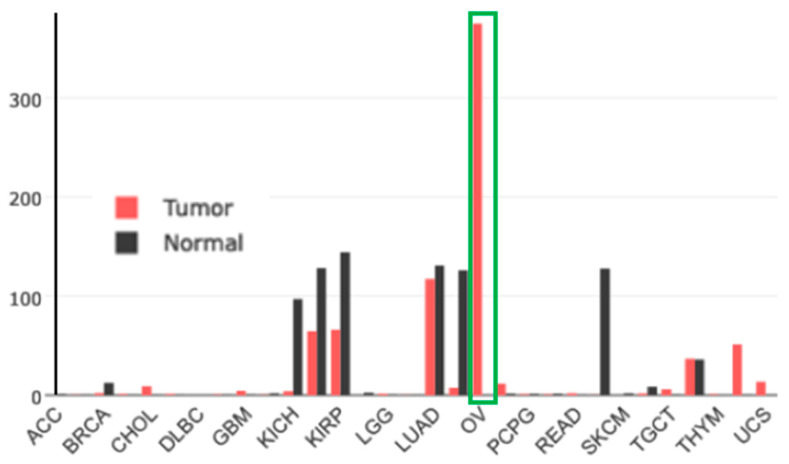
Gene-expression profile of FOLR1 in human normal and tumor tissue samples using the gene-expression profiling interactive analysis tool (http://gepia.cancer-pku.cn/ (accessed on 17 February 2023)). The cancer and normal ovarian samples are shown in the green rectangle. The height of the bar represents the median expression of the certain tumor type or normal tissue.

**Figure 14 ijms-24-07126-f014:**
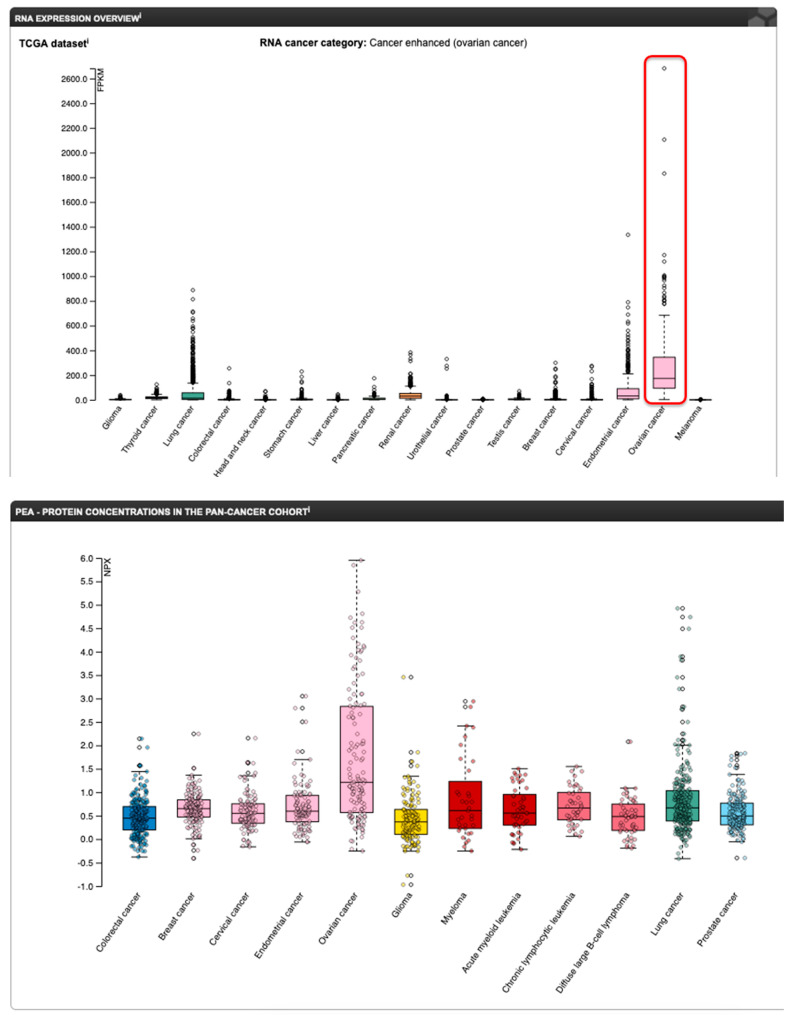
mRNA (top panel) and protein (bottom panel) expression profile of FOLR1 in various cancers (www.proteinatlas.org (accessed on 17 February 2023)). The red rectangle on the mRNA panel highlights the ovarian cancer samples.

**Figure 15 ijms-24-07126-f015:**
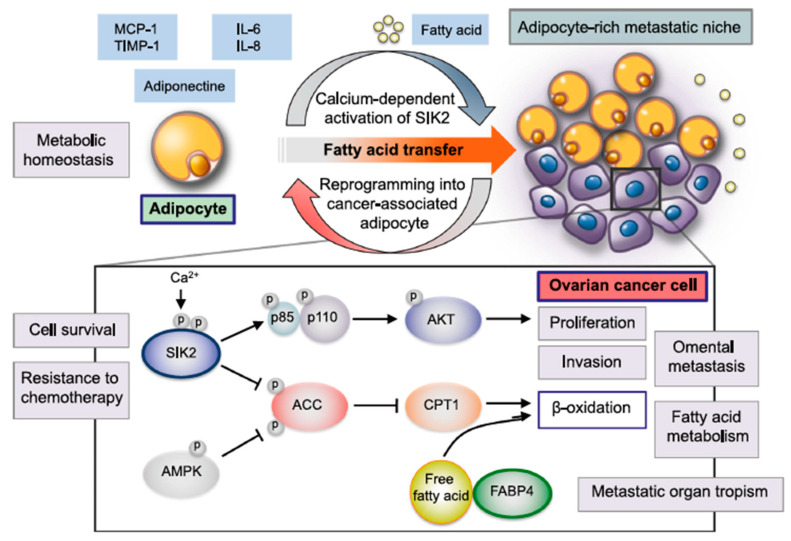
Various roles of adipocytes in the creation of the metabolic tumor microenvironment (TME) in the omentum during ovarian cancer metastasis, reprinted from Reference [92], an open-access article distributed under the terms and conditions of Creative Commons Attribution (CC BY) license (http://creativecommons.org/licenses/by/4.0/ (accessed on 17 February 2023)).

**Table 1 ijms-24-07126-t001:** Prediction and ranking of interactions between various miRNAs and clusters of metabolic genes in humans using the metaMIR Tool (http://rna.informatik.uni-freiburg.de/metaMIR/Input.jsp (accessed on 3 July 2021)).

miRNA	Final Score	Positive Combo
hsa-miR-22-3p	12.11	AKT1,BDNF,CDKN1A,CREB1,ESR1,HDAC4,HDAC6,KDM3A,KDM6B,KLF6,MAPK14,MECP2,PPARA,PPARGC1B,PRDM16,PTEN,RUNX2,SIRT1,SOD2,SP1,STAT3
hsa-miR-520c-3p	4.58	AKT1,CDKN1A,ESR1,HDAC4,KDM6B,KLF6,MECP2,PPARA,PPARGC1B,PRDM16,PTEN,RUNX2,SP1,STAT3
hsa-miR-10b-5p	4.55	BDNF,CDKN1A,CREB1,ESR1,HDAC4,KDM3A,PPARA,PPARGC1B,PRDM16,PTEN,SOD2,SP1
hsa-miR-1470	4.35	AKT1,CREB1,HDAC4,KDM6B,KLF6,MAPK14,MECP2,PPARA,PPARGC1B,PTEN,RUNX2,SOD2,STAT3
hsa-miR-5089-3p	4.29	AKT1,CREB1,KLF6,MAPK14,MECP2,PRDM16,PTEN,SIRT1,SOD2,STAT3
hsa-miR-7110-5p	4.12	CDKN1A,CREB1,KDM6B,KLF6,MECP2,PPARA,PPARGC1B,PRDM16,PTEN,RUNX2,SP1,STAT3
hsa-miR-30a-5p	4.1	AKT1,BDNF,CREB1,KDM3A,KDM6B,KLF6,PPARGC1B,PRDM16,PTEN,RUNX2,SIRT1,SOD2
hsa-miR-520a-3p	3.95	AKT1,CREB1,ESR1,HDAC4,KDM6B,KLF6,MECP2,PPARGC1B,PRDM16,PTEN,RUNX2,SOD2,STAT3
hsa-miR-520b	3.83	AKT1,ESR1,HDAC4,KDM6B,KLF6,MECP2,PPARA,PPARGC1B,PRDM16,PTEN,RUNX2,SOD2,STAT3
hsa-miR-365a-5p	3.77	CDKN1A,ESR1,HDAC4,KDM6B,KLF6,MECP2,PPARA,PPARGC1B,PRDM16,SOD2,STAT3

**Table 2 ijms-24-07126-t002:** A list of publicly available bioinformatics software used to identify candidate miRNAs and their gene targets, with particular relevance to ovarian cancer.

Tool	Web Address	Function
Target Scan Human 8.0	www.targetscan.org/vert_80 (accessed on 7 November 2022)	Search for predicted miRNA targets
metaMIR V 1.1.0	http://rna.informatik.uni-freiburg.de/metaMIR/Input.jsp (accessed on 7 November 2022)	Predict interactions between miRNAs and clusters of genes in human
OncomiR	http://www.oncomir.org/ (accessed on 7 November 2022)	WashU Pan-Cancer miRNome Atlas exploring pan-cancer microRNA dysregulation
GeneNet V 3.0.2	http://www.oncomir.org/ (accessed on 7 November 2022)	R package for learning high-dimensional dependency networks from genomic data
Cytoscape V3.7.1	https://cytoscape.org/ (accessed on 7 November 2022)	Network data integration, analysis and visualization
DiffCorr V0.4.2	https://sourceforge.net/projects/diffcorr/ (accessed on 7 November 2022)	R package to analyze differential correlations biological networks
STRING V11.5	https://string-db.org/ (accessed on 7 November 2022)	Protein–protein interaction networks Functional enrichment analysis
Webgestalt V2019	http://www.webgestalt.org/ (accessed on 7 November 2022)	Gene list over-representation analysis

**Table 3 ijms-24-07126-t003:** List of 472 proteins related to ovarian cancer with the protein–protein interaction-networks functional enrichment analysis tool String (https://string-db.org (accessed on 7 November 2022)).

ABCC3	CALR	CUL4A	FZD2	KLF9	NRXN3	RB1	STK4
ABL2	CANX	CXCL1	FZD6	KLLN	NSD1	RBBP8	STMN2
ACAP2	CARD18	CXCL10	FZD8	KRAS	NUAK1	RHOBTB3	STX17
ACO2	CASP10	CXCL11	GAB2	LATS2	OLA1	RHOC	STXBP4
ACSL4	CASP8	CXCL12	GADD45B	LHX6	OLFML3	RNF44	SUCO
ACTC1	CCL5	CXCL8	GALNT1	LIMK1	OVOL1	ROCK1	SYNCRIP
ACTR1A	CCNB1	CXCL9	GALNT14	LOX	P4HA1	ROCK2	TAGLN
ACTRT3	CCND1	CXCR3	GALR1	LPIN1	PA1	RUNX1	TAP1
ADAM12	CCND2	CYP1B1	GCNT1	LRRC15	PAK2	RUNX2	TCF21
ADAM17	CCNE1	CYTIP	GCNT2	LRRK2	PAPD7	RUNX3	TCF4
ADAM19	CCNG1	DAAM1	GCNT4	LSG1	PARP1	S1PR1	TCF7L1
ADAMDEC1	CCNG2	DCN	GCOM1	LUM	PAX7	SALL2	TEX261
ADAMTS17	CCR2	DCTN5	GCSAM	LZTS1	PCDHA10	SDC1	TGFB1
ADAMTS19	CD1D	DCX	GEMIN4	MACC1	PCDHA3	SEMA4D	THBS2
ADAMTSL1	CD2	DDB2	GFPT2	MAP2	PCDHA5	SEMA6B	TIMM17A
AGO1	CD247	DICER1	GM2A	MAP3K1	PCDHGA10	SEPTIN6	TIMMDC1
AKAP13	CD27	DKK1	GNAI3	MAP3K7	PCNA	SET	TIMP2
AKR1D1	CD38	DLG2	GPR12	MAPK1	PDCD6	SGCD	TIMP3
AKT1	CD3D	DLGAP2	GPR124	MAPK14	PDE7A	SHROOM2	TLN1
AKT2	CD3E	DNMT1	GPR83	MAPK3	PDGFRA	SIK1	TLR4
AKT3	CD44	DTD2	GRB7	MCM2	PDGFRB	SIK2	TMEM239
ALG2	CD55	DVL3	HBEGF	MED12L	PDHB	SIRT1	TMEM45A
ANKRD46	CD68	E2F2	HDGF	MET	PDZK1IP1	SIT1	TP53
ANXA8L1	CD74	E2F3	HEPHL1	MLIP	PHEX	SIX2	TP53I11
APAF1	CD82	E2F5	HEYL	MLLT3	PHLDB2	SKAP2	TRIM2
APC2	CD8A	EBF1	HIF1A	MMP10	PIEZO2	SLA2	TRIM27
ARHGAP24	CD97	EFEMP1	HLX	MMP16	PIGH	SLAMF7	TRIM31
ARHGAP28	CDC25A	EGFR	HMGA1	MMP2	PIK3CA	SLAMF8	TRIM52
ARID1A	CDC25B	EIF5A2	HMGA2	MMP9	PKP1	SLC24A4	TSC1
ARID3B	CDH1	ELAVL1	HMGB1	MSH5	PLAG1	SLC2A3	TTC14
ARL5B	CDH2	ELF5	HNRNPC	MSN	PLAU	SLC31A1	TUBB3
ASXL3	CDK1	ELN	HOXA10	MT-CO1	PLD3	SLC43A2	TWIST1
ATM	CDK12	EML1	HOXA13	MT-ND2	PLK1	SLC4A4	VAT1L
ATP5B	CDK2	EPAS1	HOXA9	MTDH	PLS3	SLC7A6	VCAN
ATR	CDK4	EPB41L3	HOXB2	MTFR1	PMAIP1	SMAD4	VEGFA
AURKB	CDK6	EPHA2	ID1	MTHFD1	POSTN	SMAD7	VEGFB
AXL	CDKN1A	EPHA4	ID4	MTSS1	POTED	SMTNL2	VEGFC
B3GNT5	CDKN2A	ERBB2	IGF1	MUC1	POU3F1	SMURF1	VIM
BAG5	CEACAM1	ERBB3	IGF1R	MUC16	PPP1R2	SMYD1	VTN
BAX	CHEK1	ERBB4	IGF2BP1	MYC	PRDM16	SNAI1	WDR17
BCL11B	CHEK2	ESRRG	IGFBPL1	MYCBP	PRKAA1	SNAI2	WNT1
BCL2	CHI3L1	FAP	IL1A	MYCN	PROX1	SOCS1	WNT5A
BCL2L1	CHST9	FAR1	IL2	MYH9	PTEN	SOCS2	WSCD1
BCR	CHSY1	FBN1	IL6R	MYO5A	PTGDR	SOD2	XIAP
BIRC5	COBLL1	FBXO28	INHBA	NEFL	PTHLH	SOS2	XXYLT1
BMF	COL11A1	FCER1G	INSR	NEFM	PTPN12	SOX11	YAP1
BMP3	COL15A1	FCRL1	ITGA5	NEUROD1	PTPN4	SOX12	YOD1
BMP4	COL1A2	FGF1	ITGB1	NEUROG1	PTPRO	SOX4	YY1
BMP7	COL3A1	FGF2	JAG1	NF1	PWWP2A	SOX9	ZEB1
BNIP3	COL5A1	FHL2	JAG2	NF2	R3HDM4	SPARC	ZEB2
BRAF	COL5A2	FN1	JAKMIP2	NFIX	RAB11FIP3	SPHK1	ZNF107
BRCA1	CPEB3	FOSL2	KCNA5	NFKB1	RAB22A	SPSB4	ZNF138
BRCA2	CPNE3	FOXA2	KDR	NHS	RAB30	SRC	ZNF181
BTLA	CRISPLD2	FOXD4L1	KEAP1	NOB1	RAB5A	SREBF1	ZNF346
C10orf128	CSF1R	FOXF2	KIAA0101	NOTCH1	RACGAP1	SREBF2	ZNF423
C11orf58	CSMD3	FOXM1	KIAA0513	NOTCH2	RAD51	SRSF1	ZNF485
C1orf105	CTGF	FOXO3	KLF12	NOTCH3	RAP1B	ST7L	ZNF521
CACNA1C	CTNNB1	FOXP1	KLF15	NREP	RARRES1	STAT3	ZNF697
CACNG8	CTSK	FUT4	KLF4	NRP1	RASD1	STK24	ZNF706

**Table 4 ijms-24-07126-t004:** List of miRNAs found to interact with 20 to 59 potential target genes.

20 to 29 Target Genes		30 to 39 Target Genes	40 to 59 Target Genes
let-7a-5p	miR-3065-5p	let-7a-3p	miR-1275
let-7b-5p	miR-30a-3p	miR-106a-5p	miR-15a-5p
let-7c-5p	miR-30d-3p	miR-106b-5p	miR-15b-5p
let-7d-5p	miR-30d-5p	miR-126-5p	miR-16-5p
let-7e-5p	miR-30e-3p	miR-129-5p	miR-195-5p
let-7f-5p	miR-30e-5p	miR-137	miR-335-3p
let-7g-5p	miR-326	miR-149-3p	miR-3607-3p
let-7i-5p	miR-330-5p	miR-17-5p	miR-373-5p
miR-105-5p	miR-340-5p	miR-181a-5p	miR-424-5p
miR-1253	miR-34a-5p	miR-181b-5p	miR-497-5p
miR-124-3p	miR-363-3p	miR-181c-5p	miR-548a-5p
miR-128-3p	miR-377-3p	miR-181d-5p	miR-548b-5p
miR-1290	miR-381-3p	miR-186-5p	miR-7-1-3p
miR-130a-5p	miR-486-3p	miR-200b-3p	miR-548d-3p
miR-130b-5p	miR-491-5p	miR-200c-3p	
miR-145-5p	miR-494-3p	miR-205-5p	
miR-182-5p	miR-506-3p	miR-20b-5p	
miR-185-5p	miR-511-5p	miR-30a-5p	
miR-1915-3p	miR-513a-3p	miR-30b-5p	
miR-200a-3p	miR-519a-3p	miR-30c-5p	
miR-204-5p	miR-519d-3p	miR-330-3p	
miR-20a-5p	miR-520b	miR-33a-3p	
miR-214-3p	miR-539-5p	miR-3688-3p	
miR-23a-3p	miR-551b-5p	miR-485-5p	
miR-23b-3p	miR-576-5p	miR-526b-3p	
miR-25-3p	miR-582-5p	miR-589-3p	
miR-26a-5p	miR-603	miR-590-3p	
miR-26b-5p	miR-629-3p	miR-93-5p	
miR-27a-3p	miR-661	miR-940	
miR-27b-3p	miR-664a-3p		
miR-29a-3p	miR-766-3p		
miR-29b-1-5p	miR-92a-3p		
miR-29b-2-5p	miR-92b-3p		
miR-29b-3p	miR-96-5p		
miR-29c-3p	miR-98-5p		
miR-3065-3p			

## Data Availability

Data presented in this article can be accessed in our previously published work [56,57] or by request to the corresponding author.

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
