# Peer review of "Strategy for Pre-Clinical Development of Active Targeting MicroRNA Oligonucleotide Therapeutics for Unmet Medical Needs"

_ijms, 2023, doi:10.3390/ijms24087126_

Round 1

Reviewer 1 Report

The authors present their modular design and development of miRNAs as therapeutic agents for the treatment of metabolic pandemics obesity, MAFLD, and ovarian cancer. The paper is logically organized and the writing style is clear. There are several good graphic presentations and the cited references seem relevant.

There are, however, a couple of things that I would like to point out:

1) The manuscript is declared as a Concept Paper. I could not find this to be an article format that IJMS excepts. Please check with the Editors.

2) In Figure 7 several of the graphs seem to be cropped/incomplete. Check the original artwork and how it is affected when converted to pdf.

3) In some Figures and Tables the text font is extremely small. The text is unreadable unless using a high magnification (see Table 1, Fig. 4, Fig. 7, Fig. 11, Fig. 15). Please enlarge the font to make it easier for the reader.

After the above points are addressed, I would recommend the publication of this manuscript in the Special Issue "The Role of microRNA in Human Diseases 2.0" of the journal IJMS.

Author Response

We would like to thank Reviewer#1 for constructive Comments and Suggestions that will certainly improve the quality of the manuscript and the ease to read it.

General Comments

  • The manuscript is declared as a Concept Paper. I could not find this to be an article format that IJMS excepts. Please check with the Editors.

Answer: Actually, it was a suggestion from the Special Issue Editor to submit the manuscript as a “Concept Paper” instead of an “Article”.

  • In Figure 7 several of the graphs seem to be cropped/incomplete. Check the original artwork and how it is affected when converted to pdf.

Answer: Indeed in Figure 7, the 4 graphs for mRNA and Protein expression levels for ASGR1 and FAT from the Human Protein Atlas Website and displayed in ‘Expression Level” mode were cropped to eliminate the tissues with little or no expression to achieve better readability. This is now mentioned in the legend of Figure 7. Furthermore, for the sake of visibility and simplicity, we removed the pictures of the liver and the human body to make more space for the mRNA and protein figures.

  • In some Figures and Tables the text font is extremely small. The text is unreadable unless using a high magnification (see Table 1, Fig. 4, Fig. 7, Fig. 11, Fig. 15). Please enlarge the font to make it easier for the reader.

Answer: As suggested, the font was enlarged to make Table 1, Fig. 4, Fig. 7, Fig. 11 and Fig. 15 easier to read. The Figures were also simplified and the number of colors minimized. Figure 11 was replaced by a new table 3.

Reviewer 2 Report

Comments to authors:

The manuscript of a Concept Paper, which was written by Drs. Marc Thibonnier and Sujoy Ghosh, is interesting, suggesting that a miR-23-3p can be a target to treat metabolic diseases. Although the presented data seem to be logically consistent, each Figure needs to be correctly drawn with appropriate descriptions on the figure legends for readers best comprehension. If figures were not drawn and explained correctly, readers cannot understand the obtained data and the conclusion neither.

Recommendation: Major revision

General comments

in vitro, in vivo, and in silico: They should be typed in italic. Some of them are already typed correctly, but some are not. Unify all through the text.

Additionally, names of genes should be typed in italic to avoid misunderstandings with that of proteins.

metabolic pandemic: metabolic diseases

ONTs could be summarized as a Table.

From page 11 to 17, analyses were carried out for the FOLR1 in ovarian cancer. Apparently, the study is not associated with any metabolic diseases. Authors had better reconsider if the part is necessary for the article, or could it be included in the other article. Otherwise, readers must be confused to read this Concept Paper.

Specific comments

Page 1, L 45: for  common; delete a space

Pages 3-4: I think this part is a background of the study, and not be included in results section.

Page 3, Figure 1: Only the essential points that are associated with this study should be shown.

Pages 3-5: How AptamiR 721 genes were obtained or selected? Authors might better draw a Ben formula or something. Indicate the biologically important meaning of KDM3A.

Why authors selected or picked up hsa-miR-22 specifically? A figure that shows the homology between the miRNA and the 3’-UTR of the KDM3A gene should be shown.

Page 5-6, Table 1: The title of the Table should not be separated but placed on the top. Explanations of the Table are described under the Table usually.

Page 7, Figure 4: Along with the expression data of the miR-22-3p, that of the KDM3A should be shown.

Page 7, Figure 8: A scale bar would be required if the sizes of the mouse are to be compared with. The feeding process should be also shown. I think readers would wonder the results (right Table). The precise obtained data indicated by values will be needed.

Page 9, Figure 7: This figure should be arranged appropriately. In the first place, illustrations of the liver and human are not necessary. What it means by NX? Some of the names are not indicated. In case it were modified from the original. Describe about that in the legend.

Page 10, Figures 8 and 9: They could be combined with to make Figure 8 a), b), c), and d) or just be summarized as a Table. Just one molecular structure of the “ss-PNA 18 mer" would be enough, delete them from each figure. Chemical formula will be a great help for comprehension. In that case, nucleotide could be shown as AGCU and the amino acids by alphabet.

Page 11-17: See General Comments.

Page 12, Table 2: Only the title should be written on the top.

Page 14, Figure 12: If the FOLR1 is included, it should be highlighted.

Page 14: Readers cannot understand the rational reason(s) why authors were moved to study FOLR1 expression in OC. Is it something to do with the miR-22-3p or other miRs?

Page 15, Figure 14: y-axis should be clearly indicated.

Page 17, Figure 16: It would better indicate some essential miRNAs and proteins, which functions or disfunctions could lead to metabolic diseases. I would never recommend authors to include biological events in ovarian cell to be adapted on an adipocyte cell. More importantly, in this study, no immunological experiments and no specific protein-phosphorylation analyses are carried out. Therefore, this figure should be completely edited or eliminated.

Other comments

Check the font and size of all through the section for References.       

Author Response

Comments and Suggestions for Authors

Comments to authors:

The manuscript of a Concept Paper, which was written by Drs. Marc Thibonnier and Sujoy Ghosh, is interesting, suggesting that a miR-23-3p can be a target to treat metabolic diseases. Although the presented data seem to be logically consistent, each Figure needs to be correctly drawn with appropriate descriptions on the figure legends for readers best comprehension. If figures were not drawn and explained correctly, readers cannot understand the obtained data and the conclusion neither.

Recommendation: Major revision

General comments

in vitro, in vivo, and in silico: They should be typed in italic. Some of them are already typed correctly, but some are not. Unify all through the text.

      Answer: All in vitro, in vivo, and in silico terms have been changed to italic format.

Additionally, names of genes should be typed in italic to avoid misunderstandings with that of proteins.

      Answer: All gene names have been italicized.

Metabolic pandemic: metabolic diseases

      Answer: We are now referring to Metabolic Pandemics instead of Metabolic Diseases throughout the manuscript.

ONTs could be summarized as a Table.

      Answer: A list of miRNA targets has been created as Table 4.

From page 11 to 17, analyses were carried out for the FOLR1 in ovarian cancer. Apparently, the study is not associated with any metabolic diseases. Authors had better reconsider if the part is necessary for the article, or could it be included in the other article. Otherwise, readers must be confused to read this Concept Paper.

      Answer: This concept paper was created to described our strategy to develop miRNA ONTs for unmet medical needs. Our most advanced program for the treatment of metabolic pandemics is extensively described in Section 2.1. We also applied our R&D strategy, knowledge and know-how of miRNA ONTs and targeted delivery to our more recent program for Ovarian Cancer as described in Section 2.2 to show that our Modular Parallel and Iterative Approach could be quickly adapted from one therapeutic area to another.

Specific comments

Page 1, L 45: for  common; delete a space

      Answer: The extra space was deleted.

Pages 3-4: I think this part is a background of the study, and not be included in results section.

      Answer: This is a good point and we moved the section on our modular strategy to the Introduction section with a simplified Figure 1.

Page 3, Figure 1: Only the essential points that are associated with this study should be shown.

            Answer: Figure 1 has been simplified and unicolored in black.

Pages 3-5: How AptamiR 721 genes were obtained or selected? Authors might better draw a Ben formula or something.

            Answer: These 721 genes were selected by using 8 publicly available in silico tools: BioCarta, Database for Annotation, Visualization and Integrated Discovery (DAVID), GeneOntology, Gene Set Enrichment Analysis (GSEA), Kyoto Encyclopedia of Genes and Genomes (KEGG), PubGene, Reactome and STRING. This information is now mentioned in section 2.1.3.

Indicate the biologically important meaning of KDM3A.

            Answer: We indicated that KDM3A regulates the expression of metabolic genes and obesity resistance (reference 54). We added a new reference showing that KDM3A senses oxygen availability to regulate PGC-1 alpha-mediated mitochondrial biogenesis (new reference 55).

Why authors selected or picked up hsa-miR-22 specifically?

            Answer: We indicated in Section 2.1.4 that various in silico tools like metaMIR were used to show that miR-22-3p is an excellent “metabolic” miRNA as shown on Table 1.

A figure that shows the homology between the miRNA and the 3’-UTR of the KDM3A gene should be shown.

            Answer: Such figure generated by TargetScan Human Release 8.0 has been added (new Figure 3).

Page 5-6, Table 1: The title of the Table should not be separated but placed on the top.

Explanations of the Table are described under the Table usually.

            Answer: The Table and related text have been reorganized and associated.

Page 7, Figure 4: Along with the expression data of the miR-22-3p, that of the KDM3A should be shown.

            Answer: The expression of KDM3A is widely distributed across human tissues and organs. This is now mentioned in the text of section 2.1.3 below the new Figure 3.

Page 7, Figure 8: A scale bar would be required if the sizes of the mouse are to be compared with. The feeding process should be also shown. I think readers would wonder the results (right Table). The precise obtained data indicated by values will be needed.

            Answer: We believe that body weight (shown on the figure) is a credible parameter of weight gain in the different groups of mice. The results shown in the table are a summary of the animal studies that were detailed in our work published in references 58 and 59.

Page 9, Figure 7: This figure should be arranged appropriately. In the first place, illustrations of the liver and human are not necessary. What it means by NX? Some of the names are not indicated. In case it were modified from the original. Describe about that in the legend.

      Answer: As suggested by the Reviewer, we removed the pictures of the liver and human body. The new figure was repositioned in the text. The tissues/organs with little or no expression of mRNAs are not shown on the pictures for sake of readability. Description of the revised figures is given in the legend.

Page 10, Figures 8 and 9: They could be combined with to make Figure 8 a), b), c), and d) or just be summarized as a Table. Just one molecular structure of the “ss-PNA 18 mer" would be enough, delete them from each figure. Chemical formula will be a great help for comprehension. In that case, nucleotide could be shown as AGCU and the amino acids by alphabet.

            Answer: Combining the 4 figures was  not legible. We decided to show only the C32 fatty acid conjugate and the hexarelin conjugate. The chemical structure of the compounds is now mentioned in the legend of the figures.

Page 11-17: See General Comments.

            Answer: See answers to General Comments.

Page 12, Table 2: Only the title should be written on the top.

            Answer: The title was repositioned.

Page 14, Figure 12: If the FOLR1 is included, it should be highlighted.

            Answer: FOLR1 is now highlighted.

Page 14: Readers cannot understand the rational reason(s) why authors were moved to study FOLR1 expression in OC. Is it something to do with the miR-22-3p or other miRs?

            Answer: The rationale for selecting FOLR1 for targeted delivery of ONTs  to primary ovarian cancer cells is now mentioned at the beginning of section 2.2.4.

Page 15, Figure 14: y-axis should be clearly indicated.

            Answer: The y-axis of the figure has been edited and the normal and tumor ovarian samples are highlighted in green.

Page 17, Figure 16: It would better indicate some essential miRNAs and proteins, which functions or disfunctions could lead to metabolic diseases. I would never recommend authors to include biological events in ovarian cell to be adapted on an adipocyte cell. More importantly, in this study, no immunological experiments and no specific protein-phosphorylation analyses are carried out. Therefore, this figure should be completely edited or eliminated.

            Answer: Indeed, we were not aware of the roles played by adipocytes in the spreading of OC and the resistance to classical treatments until we came across recent publications on the reciprocal interplay between OC cells and surrounding stromal cell types in the adipose-rich metastatic microenvironment. Therefore, we believed that our knowledge and know-how on targeted delivery to OC cells and adipocytes could be applied to treat the deadly Stage 3 and Stage 4 OCs. This could be a cancer moonshot project that we should try developing.

Other comments

Check the font and size of all through the section for References. 

            Answer: Font and size of the references were adjusted.

Round 2

Reviewer 2 Report

Comments to authors:

The manuscript of a Concept Paper, which was written by Drs. Marc Thibonnier and Sujoy Ghosh has been considerably improved comparing with the original manuscript. However, some Figures need to be edited. Before publication of the research article, I strongly recommend authors to consider revision of the manuscript.

Recommendation: Minor revision

Specific comments

Page 7, Figure 5: Indicate the meaning of arrows on the legend.

Page 8, Figure 8: On the left pictures, there are no scale bars or size markers, such as a matchbox or a candy or something. We cannot estimate size of each mouse. Therefore, each picture of the mouse would be better to be eliminated. Additionally, authors can indicate not only mean body weight but also SD value. The Table, which indicate the statistical data, is essentially required here.

Regarding the right table, if the descriptions are just summarized from references, they are enough just written on the text with appropriate references.

In summary, the Results on the right, and the left pictures should be eliminated either. Instead, authors should put a panel that shows mean weights with statistically analyzed values.

Pages 13 and 14, Table 3 1: The Table 3 should be printed on the same page.

Pages 18 and 19: Figure 15 on the pages 18 and 19 should be shown on the same page, providing A and B, respectively. If it is possible, they would better be rotated rightly.

Author Response

Thank you for your comments which allowed us to improve the quality and readability of the manuscript.

Reviewer#2 Specific comments

Page 7, Figure 5: Indicate the meaning of arrows on the legend.

      Answer: Added to the legend: Adipocyte, myocardium and skeletal muscle samples are identified by red arrows.

Page 8, Figure 8: On the left pictures, there are no scale bars or size markers, such as a matchbox or a candy or something. We cannot estimate size of each mouse. Therefore, each picture of the mouse would be better to be eliminated. Additionally, authors can indicate not only mean body weight but also SD value. The Table, which indicate the statistical data, is essentially required here.

Regarding the right table, if the descriptions are just summarized from references, they are enough just written on the text with appropriate references.

In summary, the Results on the right, and the left pictures should be eliminated either. Instead, authors should put a panel that shows mean weights with statistically analyzed values.

            Answer: We tried to modify Figure 6 as suggested, but it became too complex. Thus, as suggested by the reviewer, we eliminated Figure 6 and summarized in the text our previous findings.

Pages 13 and 14, Table 3 1: The Table 3 should be printed on the same page.

            Answer: Table 3 and its legend have been reformatted to fit on a single page. We did the same for Table 4 and its legend.

Pages 18 and 19: Figure 15 on the pages 18 and 19 should be shown on the same page, providing A and B, respectively. If it is possible, they would better be rotated rightly.

            Answer: The two panels of Figure 15 (now Figure 14) have been rotated to fit on a single page.